# Structural insight into guanylyl cyclase receptor hijacking of the kinase–Hsp90 regulatory mechanism

Nathanael A Caveney[1]*, Naotaka Tsutsumi[1,2]†, K Christopher Garcia[1,2]*

[1]Departments of Molecular and Cellular Physiology, and Structural Biology, Stanford University School of Medicine, Stanford, United States; [2]Howard Hughes Medical Institute, Stanford University School of Medicine, Stanford, United States

*For correspondence:
ncaveney@stanford.edu (NAC);
kcgarcia@stanford.edu (KCG)

Present address: †Graduate School of Medicine, Dentistry and Pharmaceutical Sciences, Okayama University, Okayama, Japan

Competing interest: The authors declare that no competing interests exist.

**Abstract** Membrane receptor guanylyl cyclases play a role in many important facets of human physiology, from regulating blood pressure to intestinal fluid secretion. The structural mechanisms which influence these important physiological processes have yet to be explored. We present the 3.9 Å resolution cryo-EM structure of the human membrane receptor guanylyl cyclase GC-C in complex with Hsp90 and its co-chaperone Cdc37, providing insight into the mechanism of Cdc37 mediated binding of GC-C to the Hsp90 regulatory complex. As a membrane protein and non-kinase client of Hsp90–Cdc37, this work shows the remarkable plasticity of Cdc37 to interact with a broad array of clients with significant sequence variation. Furthermore, this work shows how membrane receptor guanylyl cyclases hijack the regulatory mechanisms used for active kinases to facilitate their regulation. Given the known druggability of Hsp90, these insights can guide the further development of membrane receptor guanylyl cyclase-targeted therapeutics and lead to new avenues to treat hypertension, inflammatory bowel disease, and other membrane receptor guanylyl cyclase-related conditions.

## eLife assessment

In this **important** study, the human membrane receptor guanyl cyclase GC-C was expressed in hamster cells, co-purified in complex with endogenous HSP90 and CDC37 proteins, and the structure of the complex was determined by cryo-EM. The study shows that the pseudo-kinase domain of GC-C associates with CDC37 and HSP90, similarly to how the bona fide protein kinases CDK4, CRAF and BRAF have been shown to interact. The methodology used is state of the art and the evidence presented is **compelling**.

## Introduction

Cyclic guanosine monophosphate (cGMP) is an important second messenger for signaling in mammalian physiology, with roles in platelet aggregation, neurotransmission, sexual arousal, gut peristalsis, bone growth, intestinal fluid secretion, lipolysis, phototransduction, cardiac hypertrophy, oocyte maturation, and blood pressure regulation (**Potter, 2011**). Largely, cGMP is produced in response to the activation of guanylyl cyclases (GC), a class of receptors that contains both heteromeric soluble receptors ($\alpha_1$, $\alpha_2$, $\beta_1$, and $\beta_2$ in humans) and five homomeric membrane receptors (GC-A, GC-B, GC-C, GC-E, and GC-F in humans). Of note are the membrane receptor guanylyl cyclases (mGC) GC-A and GC-B, also known as natriuretic peptide receptors A and B (NPR-A and NPR-B), respectively, and GC-C, all of which have been a focus of therapeutic development. In the case of NPR-A and B, their role in regulating blood pressure in response to natriuretic peptide hormones (ANP, BNP, and

CNP) has led to the exploration of agonists for use in the treatment of cardiac failure (**Kobayashi et al., 2012**). Meanwhile, GC-C is the target of clinically approved laxative agonists, linaclotide, and plecanatide (**Miner, 2020**; **Yu and Rao, 2014**), which increase intestinal fluid secretion.

These membrane receptor GCs consist of an extracellular ligand binding domain (ECD), which acts as a conformational switch to drive intracellular rearrangements to activate the receptor (**He et al., 2001**) a transmembrane region (TM); a kinase homology domain or pseudokinase domain (PK); a dimerization domain; and a GC domain, which acts to produce cGMP. The PK domain is largely thought to be involved in scaffolding and physical transduction of the extracellular rearrangements to the GC domain, in some respects similar to the role of the PK domain in the Janus kinases of the cytokine signaling system (**Glassman et al., 2022**). In addition, the PK domains of mGCs are regulated through phosphorylation (**Potter and Garbers, 1992**; **Potter and Hunter, 1998**; **Vaandrager et al., 1993**) and via association with heat shock proteins (Hsp) (**Kumar et al., 2001**).

While the role of the phosphorylation state on mGC activity has been explored in relative detail, how the heat shock protein 90 (Hsp90) is able to regulate mGC activity is largely unknown. It has been shown that GC-A activity can be regulated through the association of Hsp90 and the co-chaperone Cdc37 (**Kumar et al., 2001**). The chaperone Cdc37 is known to assist in the Hsp90 regulation of around 60% of active kinases, both in soluble and membrane receptor form (**Taipale et al., 2012**). Given the sequence and structural similarities between the PK domains of mGCs and the active kinase domains which Hsp90–Cdc37 regulates, it is possible that mGCs have evolved to hijack the regulatory mechanisms that are more broadly deployed for active kinases.

Here, we report the 3.9 Å resolution structure of the GC-C–Hsp90–Cdc37 regulatory complex. In this structure, the core dimer of Hsp90 forms its canonical closed conformation, while Cdc37 and the C-lobe of the GC-C PK domain asymmetrically decorate the complex. The client (GC-C) is unfolded into the channel formed at the interface between the Hsp90 dimers. To our knowledge, this is the first structure of a membrane protein client of Hsp90 and the first structure of a non-kinase client of the Hsp90–Cdc37 regulatory system. This work provides a pivotal understanding of the mechanism and structural basis of kinase fold recruitment to the Hsp90–Cdc37 regulatory complex. This increased understanding can guide the further development of mGC-targeted therapeutics and lead to new avenues to treat hypertension, inflammatory bowel disease (IBD), and other mGC-related conditions. In addition, the general insights into the recruitment of Hsp90–Cdc37 clients can guide the further development of Hsp90 targeting therapeutics in cancer treatment.

## Results

### Structure of the GC-C–Hsp90–Cdc37 regulatory complex

Membrane receptor guanylyl cyclases have been largely recalcitrant to structural analysis by x-ray crystallography and electron microscopy, apart from various crystal structures of both liganded and unliganded ECDs (**He et al., 2001**; **He et al., 2006**; **Ogawa et al., 2004**; **Ogawa et al., 2010**; **van den Akker et al., 2000**). Given the relative disparity of our structural understanding, we sought to develop a stable construct to image and gain a crucial understanding of the regulatory and functional aspects of mGCs which occur intracellularly. By replacing the ligand-responsive ECD with a homodimeric leucine zipper, we mimic the ligand-activated geometry of the ECD (**He et al., 2001**), while reducing complexity of the imaged complex and increasing stability (**Figure 1A**). This complex was recombinantly expressed in mammalian cells, purified with anti-FLAG affinity chromatography, and vitrified on grids for cryo-EM analysis.

The purified sample had a substantial portion of imaged particles for which the native regulatory heat shock protein, Hsp90, and its co-chaperone, Cdc37, are bound. The *Cricetulus griseus* HSP90β and Cdc37 show remarkable sequence conservation in comparison to the human equivalents, at 99.7 and 94.2% identity, respectively. This native pulldown strategy contrasts with the structures of Hsp90–Cdc37 in complex with soluble kinases (**García-Alonso et al., 2022**; **Oberoi et al., 2022**; **Verba et al., 2016**), for which Hsp90 and Cdc37 had to be overexpressed to obtain complex suitable for imaging. Three-dimensional reconstruction of our GC-C–Hsp90–Cdc37 particles generated a 3.9 Å resolution map of the regulatory complex (**Figure 1**, **Figure 1—figure supplements 1 and 2**). A second, unsharpened map from subsequent heterogeneous refinement resolves additional density

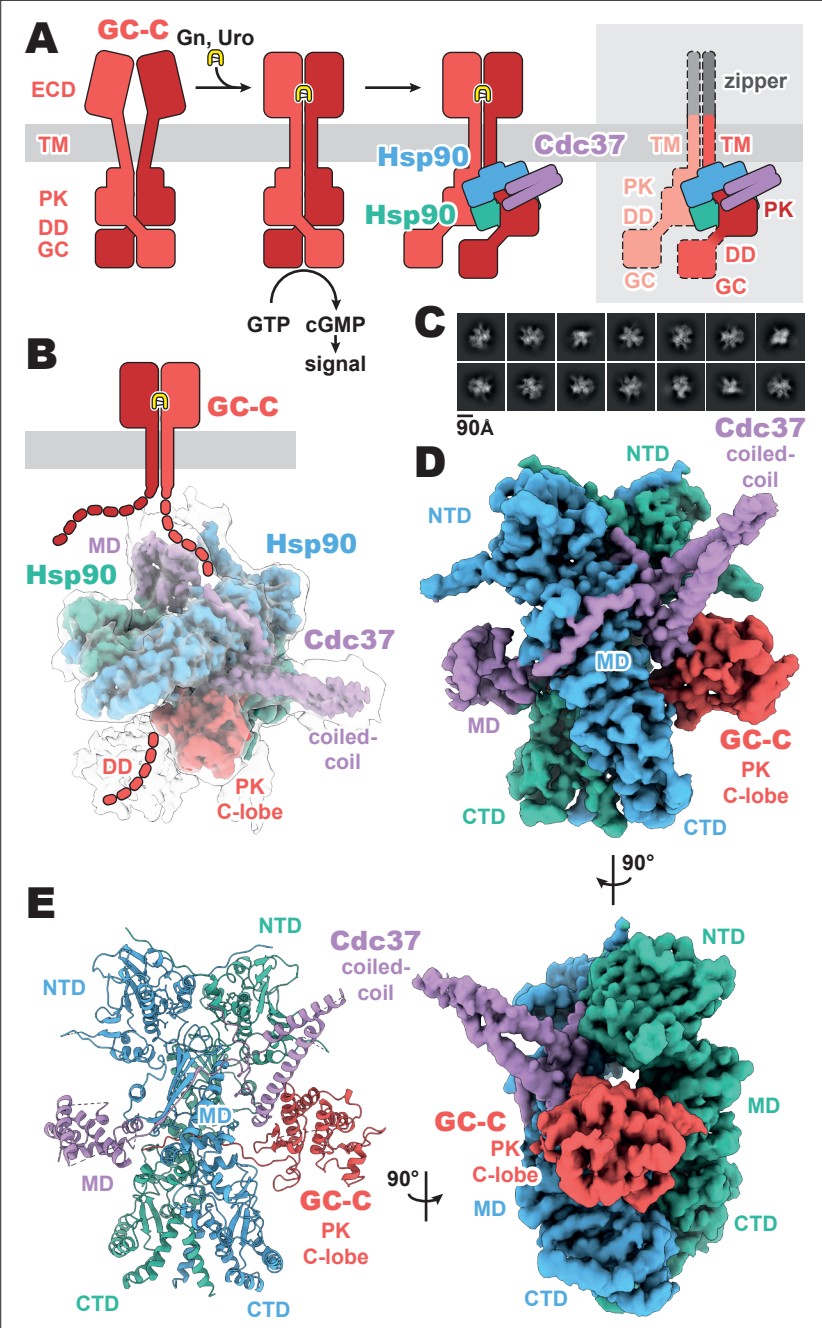

**Figure 1.** Composition and cryo-EM structure of the GC-C–Hsp90–Cdc37 regulatory complex. (**A**) Cartoon representation of the components of guanylyl cyclase C (GC-C) signaling and Hsp90–Cdc37 regulation and the zippered and activated GC-C. GC-C is colored in red, guanylin/uroguanylin (Gn/Uro) in yellow, Hsp90 in blue and teal, and Cdc37 in purple. Extracellular domains (ECD), transmembrane domain (TM), pseudokinase domain (PK), dimerization domain (DD), and guanylyl cyclase domain (GC) are labeled. In the rightmost cartoon, the regions unobserved in the cryo-EM density are in a lighter shade with a dashed outline. (**B**) The refined and sharpened cryo-EM density map of GC-C–Hsp90–Cdc37, colored as in **A**, with a transparent overlay of an unsharpened map with additional DD density resolved. Cdc37 coil-coiled and middle domain (MD) are labeled. (**C**) Reference-free 2D averages for the GC-C–Hsp90–Cdc37 complex. (**D**) The refined and sharpened cryo-EM density map of GC-C–Hsp90–Cdc37, colored as in **A** and **B**, labeled with all domains as in **A** and **B**, with the addition of Hsp90 N-terminal domain (NTD), middle domain (MD), and C-terminal domain (CTD). (**E**) Ribbon representation of a model of GC-C–Hsp90–Cdc37 complex, colored and labeled as in **A**, **B**, and **C**.

The online version of this article includes the following figure supplement(s) for figure 1:

*Figure 1 continued on next page*

*Figure 1 continued*

**Figure supplement 1.** GC-C–Hsp90–Cdc37 complex cryo-EM data processing.

**Figure supplement 2.** Representative density of GC-C–Hsp90–Cdc37.

for the dimerization domain, extending outward from the PK domain (*Figure 1B*, *Figure 1—figure supplement 1*).

The resultant GC-C–Hsp90–Cdc37 complex is a hetero-tetramer formed by one resolved monomer of the GC-C receptor bound to a dimer of Hsp90 and one Cdc37 co-chaperone (*Figure 1D*). As observed with most Hsp90–client structures, the bulk of the complex is composed of the C2 pseudo-symmetric, ATP bound, closed state Hsp90 dimer. Building on this dimeric core, the Cdc37 protrudes outward from one side with its characteristic long, coiled-coil, α-hairpin. On one face of the Hsp90 dimer core, Cdc37 interacts with the PK domain of GC-C, while an extended β-sheet wraps around to the other face, lying across and extending a β-sheet in the middle domain (MD$^{Hsp90}$) of one Hsp90 monomer. At the opposite face, the globular and α-helical Cdc37 middle domain (MD$^{Cdc37}$) is formed. The C-lobe of the GC-C PK domain packs against the N-terminal region of Cdc37 on one face of the dimeric Hsp90 core, with the N-lobe unfolding through the dimer core to interface with the MD$^{Cdc37}$ on the opposite face. N-terminal to the PK N-lobe is the TM region, the density for which was unobserved in our reconstructions. C-terminal to the PK C-lobe, we observe some poorly resolved density for the likely mobile dimerization domain in our unsharpened map. This would precede the GC domain, which is not observed in the density of our reconstructions (*Figure 1B*). Together, we can use our understanding of mGC topology and our reconstruction to orient the complex as it would sit on a membrane (*Figure 1B*), providing insight into how Hsp90 is able to access and regulate membrane protein clients. No density is observed for the second GC-C of the dimer, though it is sterically unlikely that an additional regulatory complex is forming on the second GC-C in a concurrent fashion, given the large size of the first Hsp90–Cdc37 and the requisite proximity of the second GC-C. In addition, this disruption of the native state of GC-C, as observed in our structure, would likely leave GC domains out of each other's proximity, precluding their catalytic activity while Hsp90 is bound.

## Cdc37 mediated GC-C recruitment and Hsp90 loading

Despite the recognized plasticity of Cdc37 co-chaperone binding to approximately 60% of kinases (*Taipale et al., 2012*), the importance of the Hsp90–Cdc37 complex for pseudokinase domain-containing proteins in the human proteome is not well studied. Thus, the structural basis for how Cdc37 can recruit GC-C to the Hsp90 regulatory complex is of particular interest. In our structures, we see that Cdc37 is displacing the N-lobe of the pseudokinase domain of GC-C, binding to the C-lobe at the N–C interface, and guiding the unfolded N-lobe into the Hsp90 dimer (*Figure 2*). The Cdc37–GC-C interface is relatively modest in size, with a calculated mean surface area of 689 Å$^2$ (as calculated by PISA *Krissinel and Henrick, 2007*). This interface is partly driven to form via charge complementarity, with positive contributions from a cluster of arginine residues on Cdc37 (R30, R32, R39) at the periphery of the interaction interface interacting with D609 and the polar residues Y580 and T586 (*Figure 2B*). Beyond this, the interface is likely largely driven via shape-complementarity, due to a minimal contribution from hydrogen bonding, salt-bridge formation, and aromatic packing contributions – in line with the ability of Cdc37 to chaperone such a diverse array of clients and client sequences.

As the unfolded PK N-lobe extends away from Cdc37, it enters the channel formed at the interface between the dimer of Hsp90 (*Figure 2C*). Here, GC-C residues 528–544 (VKLDTMIFGVIEYCERG) lie across the upper region of the Hsp90 CTDs, which form the floor of the channel. These CTDs form the bulk of the interaction interface as the unfolded N-lobe passes through this channel, yet there are minor contributions from the loop regions of the β-sheet from the MD$^{Hsp90}$ which extend downward into this channel region. The unfolded region is relatively poorly resolved in the density, with some reconstructions from earlier refinement having no resolvable density in this channel region – indicative of the low stability and high mobility of the unfolded N-lobe as it passes through this region.

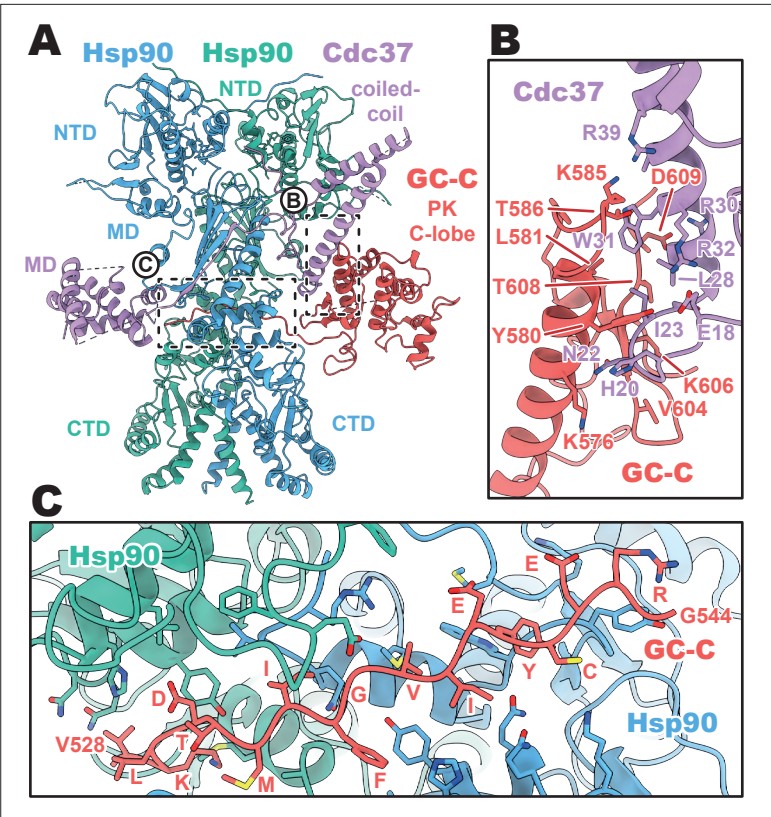

**Figure 2.** Cdc37 mediated guanylyl cyclase C (GC-C) recruitment and heat shock protein 90 (Hsp90) loading interfaces. (**A**) Ribbon representation of a model of GC-C–Hsp90–Cdc37 complex. GC-C is colored in red, Hsp90 in blue and teal, and Cdc37 in purple. Pseudokinase (PK), coil-coiled, middle (MD), C-terminal (CTD), and N-terminal (NTD) domains are labeled. (**B**) The Cdc37–GC-C interface in ribbon representation, with interacting residues drawn in sticks, colored as in **A**. (**C**) The unfolded N-lobe of GC-C PK domain as it passes between the Hsp90 dimer, in ribbon representation, with interacting residues drawn in sticks, colored as in **A** and **B**. This region's sequence is: VKLDTMIFGVIEYCERG.

The online version of this article includes the following figure supplement(s) for figure 2:

**Figure supplement 1.** Conservation of Cdc37 mediated heat shock protein 90 (Hsp90) regulation.

**Figure supplement 2.** Regulatory mechanisms for membrane receptor guanylyl cyclase (mGC) activity.

## Conservation of Cdc37 mediated Hsp90 regulation

The core structural principles of Cdc37 mediated client recruitment to Hsp90 appear to remain constant across its large range of client diversity. Across other clients–Hsp90–Cdc37 complexes with canonical soluble kinase clients (Cdk4, RAF1, B-raf) (*García-Alonso et al., 2022*; *Oberoi et al., 2022*; *Verba et al., 2016*), we see a conserved role for Cdc37 in client recruitment by associating with the C-lobe at the N-, C-lobe interface (*Figure 2—figure supplement 1A*, B). In these complexes, we see high levels of structural conservation for the Hsp90–Cdc37 (Cα RMSDs of 1.4–3.3 Å for Hsp90 and 1.5–2.5 Å for Cdc37), while the client is structurally most homogenous at the interface with Cdc37, though less structurally conserved overall (Cα RMSDs of 3.5–11.6 Å). Perhaps unsurprisingly, GC-C is one of the most divergent of these clients from a sequence perspective (*Figure 2—figure supplement 1C*), with sequence homology between the GC-C PK domain and the other client kinase domains ranging from 19 to 25% identity and 31 to 41% homology. This highlights the plasticity required of this system which can service such a vast array of clients across a broad range of sequence variations, yet more restricted fold architecture.

## Discussion

The present cryo-EM structure of GC-C–Hsp90–Cdc37 resolves the loading of GC-C, via its PK domain and interaction with Cdc37, to the Hsp90 core dimer (*Figures 1 and 2*). This complex shows significant structural similarity to the mechanism that regulates soluble active kinases (*García-Alonso et al., 2022*; *Oberoi et al., 2022*; *Verba et al., 2016*) and presumably membrane receptor kinases in the human proteome. This structural and mechanistic conservation is largely driven by the co-chaperone Cdc37, which serves as the central binding platform for these clients by associating to the fold of the kinase (or pseudokinase in the case of mGC) domain, relatively independent of sequence identity. A model whereby recruitment is largely driven by both the fold complementarity and the specific stability properties of the kinase fold has been proposed previously (*Taipale et al., 2012*). In this model, instability of a fully folded kinase domain results in partial unfolding of the C-lobe, leading Cdc37 to bind the partially unfolded state. Given the lack of functional and sequence conservation for GC-C as a client of Cdc37, our data largely fits with this model for client recruitment. It is likely that the pseudokinase domains of mGC have largely evolved to facilitate regulatory mechanisms for these receptors, both via their phosphorylation and by hijacking the regulatory mechanisms used by active soluble and membrane receptor kinases.

In the case of GC-A, previous work has shown that it associates with the Hsp90–Cdc37 complex to regulate GC activity (*Kumar et al., 2001*). The authors showed that adding geldanamycin, an Hsp90 inhibitor, reduces the overall cGMP output of cells in response to ANP stimulation while also reducing the association of the Hsp90 to GC-A. While this initially may seem counterintuitive, this data fits with a model of ligand-induced activity potentiating the instability of the PK domain, which then facilitates binding of the regulatory complex to 're-fold' GC-A for further catalysis and cGMP production – in a core regulatory complex structurally similar to that which we observe for GC-C in this work (*Figure 2—figure supplement 2*). In the case of the Hsp90 inhibitor, this would release the Hsp90 and only allow full catalytic activity for the receptor until the receptor falls into the partially unfolded state, as the Hsp90 would no longer be able to re-engage at the C-lobe when inhibited (*Figure 2—figure supplement 2*).

Interestingly there may be an additional layer of regulation involved, with crosstalk between the phosphorylation and Hsp90 regulatory mechanisms of mGC. The phosphatase PP5 is known to interact with the Hsp90–Cdc37 system and dephosphorylate Hsp90, Cdc37, and the system's kinase clients (*Oberoi et al., 2022*). PP5 has been implicated in this role for mGC (*Chinkers, 1994*), though this interaction was unable to be detected by a pull-down in a second study (*Kumar et al., 2001*). In this way, mGC association with the Hsp90–Cdc37 complex could result in multiple fates and resultant activity profiles for the receptor. When the PK of an activated mGC falls into a destabilized state, this would result in the recruitment of the Hsp90–Cdc37. First, the regulatory complex could refold the receptor to maintain the activity of the receptor (*Figure 2—figure supplement 2i*). In another scenario, the Hsp90–Cdc37 complex could additionally recruit PP5 to dephosphorylate the mGC (*Figure 2—figure supplement 2ii*). Particularly in the case of GC-A and GC-B, and to some extent GC-C (*Potter and Garbers, 1992*; *Potter and Hunter, 1998*; *Vaandrager et al., 1993*), this would impair the signaling activity of the mGC, though this could be rescued through the kinase re-association and phosphorylation. In a final scenario, the binding of the Hsp90–Cdc37 complex could result in the association of ubiquitin E3 ligases (*Schopf et al., 2017*; *Figure 2—figure supplement 2iii*), which would ubiquitinate the mGC client, leading to the removal of the receptor.

The regulation of mGC is influenced by a network of factors working in harmony to ensure proper signaling and physiological response for these important receptors. The structure of the core regulatory complex shown in this work is key to many facets of mGC regulation. We hope that the structural basis for the Hsp90 regulatory platform for mGC will drive renewed investigation into these diverse mechanisms and lead to the therapeutic manipulation of these mechanisms to improve mGC targeting therapies.

## Methods

**Key resources table**

| Reagent type (species) or resource | Designation | Source or reference | Identifiers | Additional information |
|---|---|---|---|---|
| Cell line (*Cricetulus griseus*) | Chinese hamster ovary kidney cells | GIBCO | ExpiCHO | |
| Recombinant DNA reagent | pD649-GCN4-TM-GC-C_ICD (plasmid) | This paper | | See: Methods - Cloning and protein expression |
| Software, algorithm | Data collection software | SerialEM | SerialEM | |
| Software, algorithm | Data processing software | Structura Biotechnology Inc. | cryoSPARC | |
| Software, algorithm | Data sharpening software | *Sanchez-Garcia et al., 2021* | DeepEMhancer | |
| Software, algorithm | Initial modeling software | *Jumper et al., 2021* | AlphaFold | |
| Software, algorithm | Graphics software | *Pettersen et al., 2021* | UCSF ChimeraX | |
| Software, algorithm | Modeling and refinement software | *Adams et al., 2010* | Phenix | |
| Software, algorithm | Modeling and refinement software | *Emsley and Cowtan, 2004* | Coot | |
| Software, algorithm | Model validation software | *Chen et al., 2010* | MolProbity | |

### Cloning and protein expression

For cryo-EM studies, a construct containing an HA secretion signal (MKTIIALSYIFCLVFA), a FLAG peptide (DYKDDDD), linker and 3 C cleavage site (KGSLEVLFQGPG), GCN4 homodimeric zipper (RMKQLEDKVEELLSKNYHLENEVARLKKLVGER), human GC-C regions corresponding to the small extracellular linker region, TM, and intracellular domains (residues 399–1,053), a second linker and 3 C cleavage site (AAALEVLFQGPGAA), a Protein C epitope tag (EDQVDPRLIDGK), and an 8 x His tag were cloned into a pD649 mammalian expression vector. This construct contains all domains of the native GC-C, with the exception of the ECD (*Supplementary file 1*). Protein was expressed using ExpiCHO Expression System Kit (Thermo Fisher). Briefly, ExpiCHO cells were maintained in ExpiCHO Expression Media at 37 °C with 5% $CO_2$ and gentle agitation, and transiently transfected by the expression construct and cultured according to the manufacturer's protocol. Cells were pelleted and stored at –80 °C.

### Protein purification

Cells were resuspended in 20 mM HEPES-Na pH 8.0, 300 mM NaCl, 1 mM TCEP, protease inhibitor cocktail (Sigma), and benzonase (Sigma). Cells were lysed by Dounce homogenizer and cellular debris was pelleted by low-speed centrifugation at 500 × g. Membranes were collected by centrifugation at 46,000 × g and stored at –80 °C until use. Membranes were thawed and solubilized with the addition of 1% n-dodecyl β-D-maltoside (DDM) and 0.1% cholesteryl hemisuccinate (CHS) (10:1) (Anatrace). Debris and unsolubilized membranes were pelleted by centrifugation at 46,000 × g. The supernatant was subsequently used in FLAG affinity chromatography. The supernatant was applied to M1 anti-FLAG resin. The resin was washed with 20 bed volumes of 20 mM HEPES-Na pH 8.0, 300 mM NaCl, 1 mM TCEP, 0.005% lauryl maltose neopentyl glycol (LMNG), 0.0005% CHS (10:1) (Anatrace), and 5 mM ATP. The protein complex was eluted with the addition of 200 µg/mL of FLAG peptide (DYKDDDD) (GenScript). Protein was subsequently concentrated to >2 mg/mL and used for cryo-EM imaging.

### Cryo-electron microscopy

Aliquots of 3 µL of complex were applied to glow-discharged 300 mesh UltrAuFoil (1.2/1.3) grids. The grids were blotted for 3 s at 100% humidity with an offset of 3 and plunge frozen into liquid ethane using a Vitrobot Mark IV (Thermo Fisher). Grid screening and dataset collection occurred at Stanford cEMc on a 200 kV Glacios microscope (Thermo Fisher) equipped with a K3 camera (Gatan). Movies

**Table 1.** Cryo-EM data collection, refinement, and validation statistics.

| | GC-C–Hsp90–Cdc37 complex PDB 8FX4 EMD-29523 | GC-C–Hsp90–Cdc37 complex with DD density |
|---|---|---|
| **Data collection and processing** | | |
| Nominal magnification | 45,000 | |
| Acceleration voltage (kV) | 200 | |
| Electron exposure (e⁻/Å²) ($e^-/\text{Å}^2$) | 58.8 | |
| Defocus range (μm) | 0.8–2.0 | |
| Pixel size (Å) | 0.9273 | |
| Symmetry imposed | C1 | |
| Final particle images | 165,635 | 48,283 |
| Map resolution FSC threshold | 0.143 | |
| Map resolution (Å) | 3.9 | 6.3 |
| | | |
| **Refinement** | | |
| Initial model used (PDB) | 5FWK, 7ZR5, AlphaFold | |
| Model resolution FSC threshold (Å) | 0.5 | |
| Model resolution (Å) | 4.2 | |
| Model Composition | | |
| Non-hydrogen atoms | 13,478 | |
| Protein residues | 1,654 | |
| Ligands | 2 | |
| *B*-factors (Å²) ($\text{Å}^2$) | | |
| Protein | 119.49 | |
| Ligand | 102.85 | |
| R.m.s. deviations | | |
| Bond lengths (Å) | 0.004 | |
| Bond angles (°) | 0.914 | |
| Validation | | |
| MolProbity score | 2.14 | |
| Clashscore | 13.88 | |
| Rotamer outliers (%) | 0.67 | |
| Ramachandran plot | | |
| Favored (%) | 92.0 | |
| Allowed (%) | 7.6 | |
| Outliers (%) | 0.4 | |

were collected at a magnification corresponding to a 0.9273 Å per physical pixel. The dose was set to a total of 58.8 electrons per Å². Automated data collection was carried out using SerialEM with a nominal defocus range set from –0.8 to –2.0 μM.

### Image processing

All processing was performed in cryoSPARC (*Punjani et al., 2017*) unless otherwise noted (*Figure 1—figure supplement 1*). 8788 movies were motion-corrected using patch motion correction. The contrast transfer functions (CTFs) of the flattened micrographs were determined using patch CTF and an initial stack of particles was picked using Topaz picker (*Bepler et al., 2019*). Successive rounds of reference-free 2D classification were performed to generate a particle stack of 165,635 particles. These particles were then used in ab-initio reconstruction, followed by non-uniform refinement (*Punjani et al., 2020*) and finally local refinement with a loose mask around the entire complex. This resulted in a 3.9 Å reconstruction of the GC-C–Hsp90–Cdc37 complex which was sharpened with deepEMhancer (*Sanchez-Garcia et al., 2021*). These particles were also used in a 4-class heterogeneous refinement to pull out a volume containing some resolved density for the dimerization domain of GC-C.

### Model building and refinement

The Cdk4–Hsp90β–Cdc37 (PDB 5FWK), PP5–B-Raf–Hsp90β–Cdc37 (PDB 7ZR5), and AlphaFold models for GC-C (*Jumper et al., 2021*; *Mirdita et al., 2022*) were docked into the map using UCSF Chimera X (*Pettersen et al., 2021*). A resultant hybrid model was then manually curated to contain the correct *Cricetulus griseus* sequences for Hsp90β–Cdc37 and run through Namdinator (*Kidmose et al., 2019*). This was followed by automated refinement using Phenix real space refine (*Adams et al., 2010*) and manual building in Coot (*Emsley and Cowtan, 2004*). The final model produced a favorable MolProbity score of 2.14 (*Chen et al., 2010*) with 0.4% Ramachandran outliers (*Table 1*). Model building and refinement software was installed and configured by SBGrid (*Morin et al., 2013*).

## Acknowledgements

We thank Liz Montabana and Stanford cEMc for microscope access for data collection. We thank Paul LaPointe and Kevin Jude for their insightful discussion of the Hsp90 structure and regulatory mechanisms. NAC is a CIHR postdoctoral fellow. KCG is an investigator with the Howard Hughes Medical Institute. KCG is supported by National Institutes of Health grant R01-AI51321, the Mathers Foundation, and the Ludwig Foundation.

## Additional information

### Funding

| Funder | Grant reference number | Author |
|---|---|---|
| Canadian Institutes of Health Research | Postdoctoral Fellowship | Nathanael A Caveney |
| National Institutes of Health | R01-AI51321 | K Christopher Garcia |
| Mathers Foundation | | K Christopher Garcia |
| Ludwig Foundation | | K Christopher Garcia |

The funders had no role in study design, data collection and interpretation, or the decision to submit the work for publication.

### Author contributions

Nathanael A Caveney, Conceptualization, Formal analysis, Investigation, Methodology, Writing - original draft, Writing – review and editing; Naotaka Tsutsumi, Formal analysis, Investigation, Writing – review and editing; K Christopher Garcia, Supervision, Funding acquisition, Project administration, Writing – review and editing

## Author ORCIDs

Nathanael A Caveney ⓘ http://orcid.org/0000-0003-4828-3479
Naotaka Tsutsumi ⓘ https://orcid.org/0000-0002-3617-7145
K Christopher Garcia ⓘ https://orcid.org/0000-0001-9273-0278

Reviewer #1 (Public Review): https://doi.org/10.7554/eLife.86784.3.sa1
Reviewer #2 (Public Review): https://doi.org/10.7554/eLife.86784.3.sa2
Reviewer #3 (Public Review): https://doi.org/10.7554/eLife.86784.3.sa3
Author Response https://doi.org/10.7554/eLife.86784.3.sa4

---

# Additional files

### Supplementary files

• MDAR checklist
• Supplementary file 1. Plasmids used in this study.

### Data availability

Cryo-EM maps and atomic coordinates for the GC-C-Hsp90-Cdc37 complex have been deposited in the EMDB (EMD-29523) and PDB (8FX4). Material availability: The plasmids used in this study are uploaded in (*Supplementary file 1*).

The following datasets were generated:

| Author(s) | Year | Dataset title | Dataset URL | Database and Identifier |
|---|---|---|---|---|
| Caveney NA, Garcia CK | 2023 | Cryo-EM maps and atomic coordinates for the GC-C-Hsp90-Cdc37 complex have been deposited | https://www.ebi.ac.uk/emdb/EMD-29523 | EMDataResource, EMD-29523 |
| Caveney NA, Garcia KC | 2023 | Cryo-EM maps and atomic coordinates for the GC-C-Hsp90-Cdc37 complex have been deposited | https://www.rcsb.org/structure/8FX4 | RCSB Protein Data Bank, 8FX4 |

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
