## [Editor Report · eLife assessment]

In this **important** study, the human membrane receptor guanyl cyclase GC-C was expressed in hamster cells, co-purified in complex with endogenous HSP90 and CDC37 proteins, and the structure of the complex was determined by cryo-EM. The study shows that the pseudo-kinase domain of GC-C associates with CDC37 and HSP90, similarly to how the bona fide protein kinases CDK4, CRAF and BRAF have been shown to interact. The methodology used is state of the art and the evidence presented is **compelling**.

---

## [Referee Report · Reviewer #1 (Public Review)]

Membrane receptor guanylyl cyclases are important for many physiological processes but their structures in full-length and their mechanism are poorly understood. Caveney et al. determined the cryo-EM structure of a highly engineered GC-C in a complex with endogenous HSP90 and CDC37. The structural work is solid and the structural information will be useful for the membrane receptor guanylyl cyclases field and the HSP90 field.

---

## [Referee Report · Reviewer #2 (Public Review)]

Caveney et al have overexpressed an engineered construct of the human membrane receptor guanyl cyclase GC-C in hamster cells and co-purified it with the endogenous HSP90 and CDC37. They have then determined the structure of the resultant complex by single particle cryoEM reconstruction at sufficient resolution to dock existing structures of HSP90 and CDC37, plus an AlphaFold model of the pseudo-kinase domain of the guanylyl cyclase. The novelty of the work stems from the observation that the pseudo-kinase domain of GC-C associates with CDC37 and HSP90 similarly to how the bona fide protein kinases CDK4, CRAF and BRAF have been previously shown to interact.

---

## [Referee Report · Reviewer #3 (Public Review)]

A detailed understanding of how membrane receptor guanylyl cyclases (mGC) are regulated has been hampered by the absence of structural information on the cytoplasmic regions of these signaling proteins. The study by Caveney et al. reports the 3.9Å cryo-EM structure of the human mGC cyclase, GC-C, bound to the Hsp90-Cdc37 chaperone complex. This structure represents a first view of the intracellular functional domains of any mGC and answers without doubt that Hsp90-Cdc37 recognizes mGCs via their pseudokinase (PK) domain. This is the primary breakthrough of this study. Additionally, the new structural data reveals that the manner in which Hsp90-Cdc37 recognizes the GC-C PK domain C-lobe is akin to how kinase domains of soluble kinases docks to the chaperone complex. This is the second major finding of this study, which provides a concrete framework to understand, more broadly, how Hsp90-Cdc37 recruits a large number of other diverse client proteins containing kinase or pseudokinase domains. Finally, the Hsp90-Cdc37-GC-C structure offer clues as to how GC-C may be regulated by phosphorylation and/or ubiquitinylation by serving as a platform for recruitment of PP5 and/or E3 ligases.

---

## [Author Response]

The following is the authors’ response to the original reviews.

**Reviewer #1 (Public Review):**
Membrane receptor guanylyl cyclases are important for many physiological processes but their structures in full-length and their mechanism are poorly understood. Caveney et al. determined the cryo-EM structure of a highly engineered GC-C in a complex with endogenous HSP90 and CDC37. The structural work is solid and the structural information will be useful for the membrane receptor guanylyl cyclases field and the HSP90 field. However, a detailed characterization of the protein sample is lacking. Moreover, the physiological significance of this structure is not fully exploited by supporting experiments and the mechanistic insight is currently limited.

We thank Reviewer #1 for constructive reviews and agree that this work forms the basis for future exploration by the guanylyl cyclase and HSP90 fields.

1. The characterization of the protein sample is lacking. SDS-PAGE would be useful to identify potential proteolysis, leading to the dissociation of GC dimer. Further size-exclusion chromatography would be helpful to estimate the molecular weight of the complex and to determine if only GC-C monomer is purified.

We have included a representative SDS-PAGE gel in our revised version of the manuscript (Figure 1—figure supplement 1). While we agree that SEC could be beneficial to further explore the stoichiometry of the imaged sample, we see no significant degradation of the guanylyl cyclase via SDS-PAGE, and therefore believe that the zippered construct would remain dimeric. Relatively poor yields of these samples precluded further exploration in this regard.

2. The orientation distribution of the particles is not homogenous in Fig. S1D. It would be helpful to present the 3DFSC curve to evaluate the effect of preferred orientation on the reconstruction.

While the orientational distribution is not perfectly uniform, the provided angles allowed for sufficient reconstruction of maps with no notable anisotropy. We have included 3DFSC curves in our revised version of Figure 1—figure supplement 1.

3. Description of protein expression details is lacking. Did the author use transient transfection, stable cell line or virus-mediated transduction?

We have clarified that these cells were expressed using transiently transfected ExpiCHO cells.

4. HSP90 binds ATP and is often co-purified with endogenous ATP/ADP. Is there ATP or ADP present in the sample/cryo-EM maps? Is the conformation of NBD similar to ATP-bound HSP90? The author needs to include the description/figures about the nucleotide state of HSP90.

There is clear density for present nucleotide in our reconstruction. Given the mechanistic role for ATP turnover in the release of HSP90 client (Young, Hartl, 2000 – PMID 11060043) and the resolved density, we believe the identity for this nucleotide is ATP. We have added comment to this regard in the revised manuscript: “…the C2 pseudosymmetric, ATP bound, closed state Hsp90 dimer.”

5. The catalytic domains of GC have to be dimerized to perform cyclase function. The presence of only one GC-PK monomer in the cryo-EM structure indicates the structure does not represent an active state of GC. These results suggest the GC expressed in this way is not functional. The authors need to explain why most of the GC protein is trapped in this inactive form.

Indeed, we do believe that this regulatory state is non-functional, as observed for active kinases. We have clarified this in the revised manuscript: “In addition, this disruption of the native state of GC-C, as observed in our structure, would likely leave GC domains out of each other’s proximity, precluding their catalytic activity while Hsp90 is bound.”

6. The GC-C construct used here is a highly engineered "artificial" construct, which has not been fully characterized in this work. Does this construct have similar activity as the activated wt GC-C? Does the protein (this engineered construct) expressed in CHO cells show activity?

While our original goal in developing this construct was to create an imageable construct that was locked in the active state, our current interpretation of the data is that the leucine-zipper induced, putative active geometry leads to the majority of this construct falling into the regulatory state with HSP90 binding. We make no claim to have resolved an active conformation in this work, yet believe that this state is of note due to the previously unresolved nature of these regulatory complexes for guanylyl cyclase receptors.

7. Are the residues on the interface between GC and HSP conserved in other members of membrane receptor guanylyl cyclases? Would mutations on this interface affect the activity of GC?

Given the role this structure plays in our understanding that HSP90 client recruitment is largely not driven by specific residue interactions and the ~30% identity of GC-C to NPR-A and NPR-B, we do not believe that mutations that do not significantly change the stability or fold of the PK domain would significantly modify recruitment to HSP.

8. The authors propose that targeting HSP90 would tune the activity of GC. Is there any experimental data supporting this idea?

Based on the work of Kumar et al., 2001 (PMID 11152473), we do believe that there is a functional link between HSP90 recruitment and GC activity. We hope that this work will spark further exploration of these concepts.

9. The model in Fig. S3 is largely speculative due to the lack of supporting functional data. In addition, it would be better to change the title to "structure of the protein kinase domain of guanylyl cyclase receptor in complex with HSP90 and cdc37" because the mechanistic insight is limited.

We agree that our supplemental figure is more speculative. We have referenced this in the discussion section of the manuscript and put this figure in the supplement to ensure that this is understood to be more speculative in nature.

**Reviewer #2 (Public Review):**
Caveney et al have overexpressed an engineered construct of the human membrane receptor guanyl cyclase GC-C in hamster cells and co-purified it with the endogenous HSP90 and CDC37. They have then determined the structure of the resultant complex by single particle cryoEM reconstruction at sufficient resolution to dock existing structures of HSP90 and CDC37, plus an AlphaFold model of the pseudo-kinase domain of the guanylyl cyclase. The novelty of the work stems from the observation that the pseudo-kinase domain of GC-C associates with CDC37 and HSP90 similarly to how the bona fide protein kinases CDK4, CRAF and BRAF have been previously shown to interact.The experimentation is limited to the cryoEM analysis, and is lacking additional studies that would give deeper insight into the oligomeric nature - if any - of the GC-C when bound to HSP90-CDC37 as compared to the free protein. This is relevant, as the dimerization domain downstream of the pseudokinase, is evident in the maps - albeit not well resolved - and it is not clear whether it is still able to mediate dimerization with a second free or HSP90-CDC37bound GC-C. It would also be good to see some experimentation that asks whether association with HSP90-CDC37 inhibits the guanyl cyclase activity. It is clear from previous work that HSP90-CDC37 silence the kinase activity of their bound client kinases, but in this case the catalytic guanyl cyclase is not directly associated with the chaperone complex and may still be able to function.

Given the geometry of the interaction, the dimerization domain of the GC would likely be monomerized, albeit with global dimerization remaining – contributed by the ECD, or in our case the liganded-ECD mimicking leucine zipper. Experimentally, it has been shown in live cells (Kumar et al., 2001, PMID 11152473) that the HSP90 association is required for maximal GC-A function. This is likely due to some sort of resetting nature to the associating to allow further activity, as opposed to activity during the association – given the latter is unlikely based on our structure, where the two GC domains would not be able to form the active dimerized state. Further dissection of this, while outside the scope of the current work, is of great interest.

Although the sequence alignment presented in SuppFig 2 shows that GC-C conserves the classic DFG motif that plays a critical role in the regulation of most kinases, the numbering of the sequence is absent, making it very difficult to relate this to the structural detail shown in Fig 2B. This needs to be clarified, as the interaction of CDC37-Trp31 with the DFG motifs and downstream activation loops in CRAF and BRAF have been proposed as important features of the selectivity of these kinases for the HSP90-CDC37 system, and it would be good to be able to see clearly how much of this is also conserved in the GC-C pseudokinase domain interaction. For example, is the much shorter activation segment (DFG -> APE) ordered in the complex or disordered?

We have clarified Figure 2—figure supplement 1 with additional numbering. While we agree that the DFG motif may play a role in recognition, only the first residue of this motif is interacting with CDC37 in our structure, so it may be likely that the role of this motif is more structural in maintaining a CDC37 complementary fold, as opposed to direct residue interactions. Additionally, many kinases which are not regulated by CDC37/HSP90 contain this motif. The shorter DFG->APE of GC-C is traceable with the exception of N613, S614, I615, though the density in this region reflects this loop not being well stabilized.

It was not easy to follow what was in the sample used for cryoEM. The cloning of the guanylyl cyclase (GC) component is described in the methods and they have shown some illustrations in fig 1 but a proper numbered figure of the domain organisation clearly showing domain boundaries and linker segments is really needed for a reader not familiar with the structure of GCs, especially since they have replaced the ECD with a leucine zipper in their construct. It is important to show a domain figure of what this construct looks like as well, as from the illustrations in fig 1 for examples its hard to see what's PK, DD, GC domains. It would also be helpful to see in the supplementary a gel of complex they put on the grids, to make it clearer what exactly the sample is and to reassure that the GC-C domains that are not resolved in the cryoEM are nonetheless present in the sample.

We have added in a gel figure to the supplement and clarified the content of the imaged construct in the methods section: “This construct contains all domains of the native GC-C, with the exception of the ECD.”

Overall there is only minimal proposal of mechanism or biological function based on the structure. The speculation in the Discussion of two fates - PP5 dephosphorylation or E3 ligase recruitment, is not supported by any experimentation, which is reasonable for speculation, but is also not underpinned by reference to any previously published work suggesting that these additional processes may be important. In the absence of any work by the authors can they put these speculations more in context with previously published work that supports the importance of these processes specifically for GC regulation?

We have ensured that these potential pathways only appear in the discussion section. It has been observed, for instance by Oberoi et al., 2022 that phosphatases can act on all components of a HSP90–CDC37–client system. Given there are well characterized phosphorylation sites for membrane GC receptors, we believe this is worth discussing in this manuscript, to stimulate further exploration of these mechanisms in the field. In addition, it has been reported that many E3 ligases are recruited to HSP90 complexes and can degrade rather non-specifically. It has been shown that one can generate PROTAC-like molecules to target non-specific clients to HSP90–E3 ligase machinery for degradation (Li et al., 2023). Given this proximity induced nature to E3 degradation of HSP90 clients, it would be highly likely that, at least in some cases, mGCs would be degraded by this mechanism as well.

**Reviewer #3 (Public Review):**
A detailed understanding of how membrane receptor guanylyl cyclases (mGC) are regulated has been hampered by the absence of structural information on the cytoplasmic regions of these signaling proteins. The study by Caveney et al. reports the 3.9Å cryo-EM structure of the human mGC cyclase, GC-C, bound to the Hsp90-Cdc37 chaperone complex. This structure represents a first view of the intracellular functional domains of any mGC and answers without doubt that Hsp90-Cdc37 recognizes mGCs via their pseudokinase (PK) domain. This is the primary breakthrough of this study. Additionally, the new structural data reveals that the manner in which Hsp90-Cdc37 recognizes the GC-C PK domain C-lobe is akin to how kinase domains of soluble kinases docks to the chaperone complex. This is the second major finding of this study, which provides a concrete framework to understand, more broadly, how Hsp90-Cdc37 recruits a large number of other diverse client proteins containing kinase or pseudokinase domains. Finally, the Hsp90-Cdc37-GC-C structure offer clues as to how GC-C may be regulated by phosphorylation and/or ubiquitinylation by serving as a platform for recruitment of PP5 and/or E3 ligases.Comments:1. The authors used an interesting approach to obtain the GC-C-Hsp90-Cdc37 complex. Flagtagged human GC-C was overexpressed in CHO cells with the expectation of co-purifying endogenous hamster homologs of Hsp90 and Cdc37. There are several points worth noting:a) It is not clear from the data presented (Figure 1C, Suppl Fig 1A) or the Methods the percentage of particles in the cryo-EM specimen that represent the GC-C-Hsp90-Cdc37 complex. Presumably, some fraction of GC-C isolated will not be associated with Hsp90Cdc37. If a very large portion of GC-C is associated with Hsp90-Cdc37, it would be good to explain why this is to be expected. Are 2D/3D classes corresponding to the activated GC-C dimer found? If not, why?

While we see some traces of GC-C not bound by Hsp90, there is, in the least, a significant alignment bias for the Hsp90 bound complex. We believe that the engineered construct, which we designed to be locked in a putative active conformation, is going through catalytic cycles to some point where the regulatory mechanism is kicking in. It may be that for proper resetting of the receptor, the receptor needs to cycle back through an unliganded, inactive conformation, which our leucine zipper construct is unable to allow, thus locking our GC in the regulatory complex, though this is speculation.

b) Figure 1A suggests that GC-C is phosphorylated before recruitment of Hsp90-Cdc37. What is the phosphorylation status of the GC-C specimen that was imaged by cryo-EM?

We had placed the P in grey in this figure to represent the potential for the active state to be phosphorylated. For GC-C in particular, the phosphorylation state does not affect activity as much as GC-A and GC-B for example. We have removed this P from the figure for clarity.

c) The resolution of the cryo-EM map (3.9 Å) is too low for unambiguous identification of proteins. Please provide more precise justification for the claim that the densities observed do in fact correspond to hamster Hsp90 and Cdc37.

While we agree that the resolution is limiting for protein identification, the fact that we are using a very stringent FLAG purification allows confidence in the ID for our target, GC-C. For Hsp90 and Cdc37, we are confident that they are endogenous hamster Hsp90 and Cdc37, given the large structural similarity observed in comparison to prior Hsp90/Cdc37/client complex structures, and the ID/register well confirmed by the placement of bulky residues.

d) The authors state that human GC-C pulls down hamster Hsp90-cdc37 but soluble kinases cannot, despite the high sequence identity between human and hamster Hsp90-cdc37. Is this because GC-C recognition is more promiscuous? Can this difference be understood in light of the new structural information presented?

“This native pulldown strategy contrasts with the structures of Hsp90–Cdc37 in complex with soluble kinases (García-Alonso et al., 2022; Oberoi et al., 2022; Verba et al., 2016), for which Hsp90 and Cdc37 had to be overexpressed to obtain complex suitable for imaging.”

It is our understanding, from reading the papers cited above, that Hsp90/Cdc37 needed to be overexpressed to obtain these samples for imaging. We use a different strategy because our sample does not require overexpression of Hsp90 and Cdc37. This may be because of something specific to hamster cells, which were (presumably) not tested in the above studies, or it could be something specific to do with GC-C.

2. A large portion of the enforced GC-C dimer was not visible in the cryo-EM maps. It is not easy to learn from Figure 1 exactly which parts of the GC-C construct was sufficiently ordered and observed structurally. Please improve Figure 1.

We have adjusted Figure 1 to better depict what is observed in the cryoEM density.

3. On page 4, the authors claim that they are able to orient the GC-C-Hsp90-Cdc37 complex "as it would sit on a membrane" and referred to Figure 1B. It is not clear what is implied here. Does Hsp90-Cdc37 binding constrain the complex to face the inner leaflet of the membrane in a specific orientation as shown in Figure 1B? If true, this could potentially have important functional implications. Please illustrate how this was deduced based on the information available.

Given the observed density for the PK domain, which is membrane proximal, we can safely assume that the TM would be located immediately above this region. Given the size of Hsp90 and assuming the soluble Hsp90 must sit below the membrane, we can determine, with some accuracy the relative orientation of this complex next to the membrane. This orientation is depicted in Figure 1B.

4. Also on page 4, it is stated that it is sterically unlikely an additional Hsp90-Cdc37 complex is associated with the other copy of GC-C in the leucine zippered dimer. It is not obvious to the reader how this may be the case. An additional figure could help make this more clear. Additional biochemical evidence will also help. The absence of GC-C-Hsp90-Cdc37 dimers in cryo-EM micrographs can also support the argument.

We have clarified this: “is sterically unlikely that an additional regulatory complex is forming on the second GC-C in a concurrent fashion, given the large size of the first Hsp90–Cdc37 and the requisite proximity of the second GC-C.”

5. Some comments on Figure 2:a) NTD and CTD are mislabeled in Figure 2A.

Thank you for catching this, we have fixed this.

b) The authors should show cryo-EM density to support their modeling of GC-C in Figures 2B and C.

We have provided maps and models to the reviewer and will release these maps and models upon publication so that all relevant densities can be interpreted to their fullest extent by readers. In addition, we have added representative density panels to Figure 1-figure supplement 2.

6. The authors claim that Hsp90-Cdc37 clients are more similar structurally near the cdc37 interface. Please illustrate this with additional figures. Suppl. Figure 2 is inadequate for this purpose.

We have added a structural overlay to Figure 2—figure supplement 1A to illustrate this.

The authors can also consider adding a more detailed discussion comparing the interactions between the pseudokinase/kinase C-lobe and Cdc37 in known structures. Is shape/charge complementarity a universal feature of cdc37-dependent kinase/pseudokinase recruitment? It would be interesting to also consider if it would be possible to predict which of the ~60 human pseudokinases are possible Hsp90-Cdc37 clients. New structural findings from this study and publicly available AI-predicted protein structures could help.

While the use of AI to predict pseudokinase interactions would indeed be interesting, we believe this is outside the scope of this work. Given methodology is in place for determination of kinase clients for Hsp90 (Taipale et al., 2012), this could be an additional route to obtain this information in future work.

**Reviewer #2 (Recommendations For The Authors):**
In Figure 1B the authors show a large unaccounted-for region of density which they speculate may be due to the dimerization domain. That this is lost in the sharpened maps suggests that it is more mobile than the core which probably dominates the automatic mask generation used by cryoSPARC. It would be very interesting to try and resolve this region further by using focussed classification and refinement - probably in RELION. This would add further novelty, as so far in the three HSP90-CDC37 kinase complexes previously described, little is seen outside the C-terminal lobe of the kinase (or in this case pseudokinase) lobe.

Given the structurally uncharacterized nature of the DD and GC domains for mGCs, using computational means to further our understanding of these regions was attempted. Across several software packages, these attempts were unsuccessful. We will be uploading these micrographs to EMPIAR shortly after publication, which will allow for other groups to re-process this data as they see fit and as new software techniques emerge in this rapidly developing field. We believe that the partially unfolded nature of the PK domain is providing too much of a hinge point prior to the DD for the software to be able to resolve this currently.